# Why Humans Prefer Phylogenetically Closer Species: An Evolutionary, Neurocognitive, and Cultural Synthesis

**DOI:** 10.3390/biology14101438

**Published:** 2025-10-18

**Authors:** Antonio Ragusa

**Affiliations:** Obstetrics and Gynecology Unit, Sassuolo Hospital, 41049 Sassuolo, Italy; antonio.ragusa@gmail.com

**Keywords:** empathy, affective preference, phylogeny, domestication, oxytocin, predictive processing, conservation psychology, animal welfare

## Abstract

Humans often feel more empathy for animals that are evolutionarily closer to us, such as dogs, cats, or horses, than for more distant species like reptiles, fish, or insects. This difference arises from how our brains recognize signals of emotion and intention—faces, eyes, voices, and movements that resemble our own. Shared biology, particularly the hormones that support bonding and care, strengthens these emotional ties. Culture and education also play a role: children’s stories, pets at home, and media images usually focus on mammals, while other animals are less visible or portrayed as dangerous. Yet, this bias can change. When people learn about the intelligence of octopuses, the cooperation of bees, or the parental care of fish, they often feel greater compassion and respect. Understanding why empathy follows evolutionary proximity helps educators and conservationists design ways to extend moral concern to all living beings. Protecting biodiversity then becomes not only a scientific or ethical goal, but also an act of empathy that connects human well-being with the health of the planet.

## 1. Introduction

Humans readily form attachments to nonhuman animals, yet these attachments are not evenly distributed across the tree of life. Dogs, cats, horses, and great apes typically elicit warm affect; snakes, insects, or fish less so (Figure 1). Although individual exceptions are common, the population-level pattern is consistent in surveys, laboratory experiments, and behavioral choices (e.g., donations, adoption intentions, rescue priorities). This paper proposes a multi-level framework that treats empathy and preference as products of (i) evolutionary proximity and signal recognizability, (ii) neurocognitive architecture tuned for mammalian social cues, (iii) domestication and niche construction, and (iv) cultural learning and moral norms. Because empathy is partly predictive—an inference over internal states from perceptual cues—species that emit human-familiar signals are easier to model, and thus easier to care about.

## 2. Defining Terms and Scope

Empathy refers to a set of processes—affective sharing, cognitive perspective taking, and empathic concern—that allow an observer to represent and respond to another’s state. Affective preference denotes the motivational valence to approach, protect, or affiliate. Phylogenetic distance indicates evolutionary relatedness; however, for perception it often proxies morphological and behavioral similarity. The present account focuses on ordinary human observers without specialized training and on nonhuman animals encountered in daily life, media, or education. Differences due to professional expertise (e.g., herpetologists), phobias, or culturally specific practices are addressed as moderators.

## 3. The Evolutionary Proximity Hypothesis

A core claim is that phylogenetic proximity increases the overlap between human priors and animal signals. Mammals share homologous facial musculature, vocal tracts within familiar frequency ranges, and body plans with bilateral limbs and forward-facing heads. These features make their movements predictable to a brain optimized for conspecific interaction. In predictive-processing [1,2] terms, the human brain maintains generative models of agents; when observed cues fit these models, prediction error is low and empathy is facilitated. Reptiles and arthropods violate many expectations (e.g., leg number, postures, facial expressiveness), inflating prediction error and reducing intuitive mind perception.

## 4. Perception and Signal Recognizability

Face, voice, and biological-motion systems matured under selection pressures to decode conspecific signals. The fusiform gyrus, superior temporal sulcus, and temporal voice areas respond strongly to faces, eyes, gaze shifts, and articulated motion. Mammalian faces provide sclera visibility, eyebrow movement, and orofacial cues that humans read effortlessly. By contrast, most fish and reptiles lack mobile facial musculature; insects signal through antennae, pheromones, or polarized light. Because these channels are not privileged in human perception, the mapping from stimulus to internal state is weak, lowering empathic accuracy. Perceptual familiarity also affects aesthetics: round eyes, neotenous proportions, and rhythmic gait appear ‘cute’ or pleasing because they match stable priors.

## 5. Neurocognitive Architecture for Social Resonance

Humans possess neural systems that resonate with observed actions and emotions. Motor resonance—often described via mirror mechanisms [3]—supports action understanding by mapping another’s movement onto the observer’s motor programs. Affective resonance engages limbic and salience networks (amygdala, anterior insula, anterior cingulate). These systems are more easily driven by movement kinematics and facial configurations that align with human templates, which mammals disproportionately supply. Functional neuroimaging shows stronger coupling between perceptual and affective circuitry when participants view familiar mammalian interactions (e.g., grooming, play) compared to low-familiarity taxa. Neuroendocrine modulators such as oxytocin [4] and dopamine further potentiate affiliative responses; in dog–human dyads, mutual gaze and friendly touch can elevate oxytocin [4] in both species, reinforcing bonding and prosocial motivation.

## 6. Developmental Trajectories and Learning

Ontogeny provides a natural experiment. Infants preferentially attend to faces and biological motion, track gaze, and imitate mouth movements—capacities that scaffold social learning. As children encounter animals, mammals provide rich face-like cues and contingent behavior, accelerating category learning and empathy calibration. Cultural scripts quickly layer on: storybooks and media often centralize mammals with human-like voices and intentions, further reinforcing the mapping between mammalian cues and mental states. At the same time, predispositions to detect snakes and spiders rapidly (a potential evolutionary defense) can bias affect downward for those taxa, especially without countervailing positive experiences. Early exposure matters: children raised with reptiles or in insect-friendly environments often display attenuated gradients [5,6,7].

## 7. Domestication, Coevolution, and Niche Construction

Domestication is both biological and cultural evolution. Selective breeding favored traits that increase inter-species tolerance and communication—attention to human gaze, sensitivity to pointing, social reward from interaction, and juvenile morphology (neoteny). Dogs exemplify this process: they track human referential gestures, form attachment bonds, and recruit oxytocinergic pathways during affiliative exchange [4]. Nagasawa et al. [4] show that such gaze-mediated bonding also exists between us and our closest animal companions, dogs, mutual gazing increased oxytocin levels, and sniffing oxytocin increased gazing in dogs, an effect that transferred to their owners. Horses and cats show complementary profiles. From the human side, living in multispecies households shapes norms, language (‘fur babies’), and caregiving routines, which generalize to other mammals through analogical reasoning. The domestication loop thus strengthens empathy for mammals generically, not only for pets. The involvement of the mediator oxytocin in transspecies empathic phenomena is very important. Tusar Giri et al. studied a rat model for labor induction with oxytocin [8]. The authors demonstrated that oxytocin exposure at birth altered early life communication in the pups and oxytocin exposed offspring had sex-specific deficits in empathy and brain connectivity. For every 500 mIU increase in the cumulative dose of oxytocin or 500 min increase in the duration of oxytocin exposure, the odds of autism spectrum disorders (ASDs) were increased by 1.1 or 2.1, respectively, among males; these associations were absent in female offspring. Ultimately, oxytocin leads to sex-specific disruption of oxytocinergic signaling in the developing brain, decreased communicative ability of pups, and reduced empathy-like behaviors.

## 8. Cultural Norms, Moral Circles, and Media

Culture powerfully modulates the gradient. Dietary practices, taboos, and religious values can elevate or suppress empathy for specific taxa (e.g., sacred cattle, revered elephants, protected raptors). Urbanization reduces routine contact with non-mammalian fauna, potentially narrowing people’s experiential base. Media dynamics privilege charismatic megafauna [9]—large mammals with expressive faces—amplifying attention and donations. Anthropomorphic storytelling simplifies mind attribution for mammals but not for insects. Educational systems can counterbalance these biases by emphasizing ecological roles of less-loved taxa (pollination by bees, nutrient cycling by detritivores) [10,11,12,13]. Anthropomorphic storytelling facilitates children’s attribution of intentions, emotions, and mental states to mammals, whose morphology and behavior already resemble humans. By contrast, insects and other invertebrates do not benefit from such narrative scaffolding: their non-mammalian body plans and communication systems remain opaque to naïve observers, even when described in human-like terms [12]. This asymmetry reinforces phylogenetic empathy gradients, privileging charismatic vertebrates in early education. Yet, educational systems can actively counterbalance these biases by highlighting the ecological indispensability of less-loved taxa. Instructional strategies that emphasize pollination by bees, decomposition and nutrient cycling by detritivores, or the role of insects as keystone species have been shown to foster conservation awareness and more positive attitudes toward these animals [11]. Integrating accurate biological information with experiential learning—such as outdoor exploration, school gardens, or direct observation—supports a more balanced moral regard across species.

## 9. Boundary Cases and Exceptions

Despite the overall phylogenetic gradient in human empathy, certain “exceptions” highlight the specific features that drive mind perception across taxa. Cephalopods, for instance, consistently elicit fascination and moral concern that exceed their phylogenetic distance. Their problem-solving abilities, tool use, play-like behaviors, and highly flexible body-language displays provide salient cues of intentionality and agency [12]. Such traits map onto human social-cognitive templates, enabling people to attribute intelligence and even personality to octopuses and cuttlefish.

Similarly, cetaceans and parrots—though phylogenetically distant from primates—demonstrate advanced vocal learning, social traditions, and cultural transmission. Dolphins and whales maintain cooperative alliances and exhibit helping behaviors that resemble empathy, while parrots engage in sophisticated vocal mimicry and social play. These convergences in communication and prosociality align with human capacities for language and culture, enhancing perceived kinship and elevating moral concern [13,14].

Conversely, not all mammals are perceived as equally empathogenic. Bats, hyenas, and rats often evoke aversion due to nocturnality, scavenging behavior, or entrenched cultural narratives that cast them as sinister, dirty, or dangerous [15]. These exceptions are not random noise in the empathy gradient; rather, they reveal the cognitive features most critical for mind attribution. Specifically, cognitive complexity, sociality, and communicative transparency emerge as the strongest predictors of perceived intelligence and moral standing—often overriding simple phylogenetic proximity.

Proximity matters because it reduces perceptual and inferential distance: mammalian faces, voices, and motions are easier for human brains to parse; social rewards are amplified by conserved neuroendocrine systems; history and culture have built niches around mammal–human cooperation. Yet, proximity is not destiny: when observers witness agency, care, or intelligence in distant taxa—octopus exploration, bee social dances, reptile parental defense—the moral circle expands. Designing environments and messages that reveal these features is a tractable route to more inclusive empathy.

## 10. Measuring Empathy and Preference

Assessment of human empathy toward nonhuman animals increasingly employs triangulation across multiple methodologies (Figure 2) each capturing different layers of the evaluative process. Self-report instruments such as Likert-type scales assess explicit attitudes, including liking, willingness to protect, and perceived sentience. These measures are efficient but subject to social desirability biases and anthropocentric framing [16].

Complementing them, behavioral proxies provide more ecologically valid insights. Donation paradigms, petition signatures, and resource-allocation tasks reveal how individuals concretely prioritize species when prosocial resources are limited. Experimental designs using economic trade-offs demonstrate that participants frequently allocate more to charismatic mammals but that preferences can be reshaped when ecological functions of less-favored taxa are made salient [17].

Psychophysiological indices add an additional layer by quantifying embodied responses. Measures such as heart-rate variability, skin conductance, and pupil dilation reliably differentiate reactions to animal distress, often scaling with phylogenetic proximity and perceived similarity [18]. Eye-tracking paradigms further refine this approach by showing that humans preferentially fixate on mammalian faces, especially eyes, and on injury sites, whereas reptiles or invertebrates elicit shorter or more diffuse fixations [19].

Neuroimaging provides a complementary neural correlate. Functional MRI studies have reported stronger activation in empathy-related networks—such as the anterior insula and anterior cingulate cortex—when participants observe suffering in mammals compared to reptiles or insects [20]. Together, these converging methodologies reveal a reliable but malleable empathy gradient: while baseline preferences typically privilege mammals, experimental manipulations that emphasize ecological functions, cognitive capacities, or social behaviors of less-preferred taxa can recalibrate responses within a single session. Such findings underscore the importance of integrating explicit, behavioral, physiological, and neural measures to capture both the stability and plasticity of human–animal empathy.

## 11. Predictive Processing and Computational Models

Formalizing the phenomenon clarifies mechanism. In a predictive processing [1,2] framework, the brain minimizes prediction error by updating beliefs about an observed agent. When species share motion priors and expressive cues with humans, less updating is required to infer internal states; empathy can be computed quickly and confidently. For distant taxa, priors are mismatched; uncertainty and error remain high, dampening empathic concern. Computational models that encode feature overlap (e.g., face mobility, vocal formants, limb articulation) and social priors (group living, parental care) reproduce human ratings of sentience and protection priority. These models also predict exception cases by weighting intelligence and social learning heavily.

## 12. Survival-Related Biases and Risk Management

From an adaptive perspective, the tendency to preferentially empathize with some species over others can be understood as the product of evolutionary pressures that have improved survival and reproductive success. This bias likely stems from the coevolutionary relationships humans have developed with specific animals that have offered tangible benefits, such as social mammals (e.g., dogs, primates) that act as allies by alerting humans to danger [21], assisting in hunting [22], or facilitating transportation [23]. These alliances not only provided direct benefits but also strengthened empathic behaviors through positive reinforcement and social bonding mechanisms.

Conversely, humans evolved to avoid species that posed direct threats, such as venomous animals (snakes, spiders) or disease-transmitting species (mosquitoes, rodents), because such avoidance strategies significantly increased the likelihood of survival [24]. These innate biases exemplify how evolved cognitive heuristics functioned as adaptive tools, shaping responses in ways that favored conservation.

Modern environments, however, have radically altered the ecological contingencies that shaped these biases. Urbanization and technological advances have reduced the frequency and severity of encounters with certain species, thereby attenuating or even reversing these adaptive responses. For example, species that were once widely avoided due to their potential for harm may now be perceived differently due to changes in habitat and human perception, often influenced by the media or cultural narratives [25]. Despite these changes, cognitive biases tend to persist because they are deeply rooted in our neurocognitive architecture, leading to persistent tendencies toward overgeneralized behaviors.

Dogs versus cats illustrate different channels to human empathy: dogs [26] excel at joint attention and following points, which map directly onto human communicative priors; cats often recruit touch and proximity routines that modulate parasympathetic tone. Horses, large prey animals, exhibit exquisitely readable ear and head cues that riders learn to interpret; empathy flows through trained sensorimotor contingencies. In contrast, snakes’ locomotion violates bipedal/quadrupedal priors, and their minimal facial expressivity undermines mind perception despite sophisticated chemosensory worlds hidden from humans. Bees—key pollinators—trigger mixed responses: fear in some, fascination in others; targeted education that links bee behavior to food security reliably elevates concern. Cephalopods, with their distributed nervous systems and dynamic body patterning, achieve high perceived intelligence when humans are shown evidence of problem solving, play, or individual temperaments.

Distinguishing between vestigial biases (i.e., remnants of evolved responses that no longer serve a current adaptive function) and those that remain environmentally appropriate is critical for effective risk communication and humane pest management. For example, recognizing that fear of snakes may be an evolved bias can inform educational campaigns aimed at reducing unnecessary panic while maintaining due caution. Conversely, biases that no longer serve a protective function can hinder coexistence and

Understanding these evolutionary and cognitive underpinnings informs strategies for public health communication, animal conservation efforts, and pest management practices. Aligning interventions with underlying biases, addressing them, or mitigating them, is key.

## 13. Health and Well-Being Correlates in Humans

Interactions with companion animals are increasingly recognized as a biobehavioral interface with measurable consequences for human health and well-being. Beyond anecdotal reports, controlled studies suggest that affiliative contact with dogs, cats, and other companion species can downregulate hypothalamic–pituitary–adrenal (HPA) axis activity, as reflected in lower cortisol responses during stress-induction paradigms [27]. Physical touch—such as stroking or grooming—has been associated with parasympathetic activation indexed by heart-rate variability, oxytocin release, and attenuated sympathetic arousal, paralleling mechanisms observed in mammalian allogrooming and maternal care [28]. In social-cognitive terms, companion animals act as “social catalysts”, increasing interpersonal interactions and perceived social support, thereby mitigating loneliness and social exclusion in vulnerable groups, including older adults and adolescents [29,30].

At the same time, the robustness of these effects varies. Meta-analytic reviews highlight substantial heterogeneity, with small-to-moderate effects on cardiovascular endpoints (e.g., blood pressure, survival post–myocardial infarction) and inconsistent findings on mental health outcomes [31]. Methodological challenges include disentangling causal pathways from self-selection biases—since individuals who choose to own pets may differ systematically in personality, socioeconomic status, or baseline health [32]. Longitudinal and randomized controlled designs remain rare, but natural experiments, such as pet adoption or bereavement, provide valuable quasi-experimental insights [33].

From an evolutionary–comparative perspective, interspecies affiliation may exploit conserved mammalian circuits underlying social bonding and stress regulation. Neural substrates implicated in conspecific attachment—including the oxytocinergic system and mesolimbic reward pathways—are engaged by interactions with companion animals, suggesting partial phylogenetic generalization of affiliative mechanisms [4]. Thus, companion-animal interactions may not simply be cultural artifacts but reflect deeper evolutionary continuities, aligning with the phylogenetic-proximity hypothesis that empathy and affiliative concern are preferentially extended to taxa sharing mammalian socio-emotional repertoires.

## 14. Ethical Implications and Conservation

A proximity-driven empathy gradient shapes human moral concern, with phylogenetically closer, mammalian species disproportionately attracting conservation attention and welfare protections. This “charisma bias” channels resources toward large vertebrates—so-called “flagship” or “charismatic megafauna”—while taxa with equivalent or greater ecological importance, such as insects, amphibians, and detritivores, remain marginalized [34,35]. Such asymmetry is not merely symbolic: funding analyses show that conservation budgets are consistently skewed toward mammals and birds, often independent of extinction risk or ecological role [36,37]. The result is a potential misallocation of scarce resources that undermines biodiversity protection at the ecosystem level (Figure 3).

Ethical frameworks that extend moral consideration beyond phylogenetic proximity—by weighing sentience, capacity for suffering, and ecological function—offer counterbalances to anthropocentric intuitions [38,39]. Utilitarian and biocentric approaches emphasize that the welfare of less charismatic taxa, including invertebrates, merits systematic evaluation, not least because of their essential ecosystem services, such as pollination, nutrient cycling, and trophic regulation [40].

From a practical standpoint, communication strategies can broaden public concern by highlighting traits in under-loved species that align with human moral intuitions. For example, demonstrating problem-solving in corvids, parental care in fish, or cooperative behavior in social insects can foster empathy and increase willingness to protect [41,42]. Importantly, such messaging need not deny human affinities for mammals; rather, it reframes the narrative to recognize convergent capacities across taxa that resonate with human cognitive biases. In doing so, it bridges evolutionary continuity with ethical inclusivity, ensuring that conservation decisions reflect both ecological necessity and fairness in moral scope.

## 15. Education and Communication Strategies

Interventions can play a pivotal role in reshaping human preferences and attenuating taxonomic empathy gradients (Table 1). Experimental work shows that even brief educational modules highlighting both the instrumental (e.g., ecosystem services such as pollination, nutrient cycling, or pest control) and intrinsic value of reptiles, fish, and invertebrates can significantly increase willingness to protect them [43]. Citizen-science initiatives, particularly those that involve tactile engagement such as handling invertebrates or observing reptiles in naturalistic settings, reduce disgust responses and fear conditioning while simultaneously enhancing ecological knowledge and stewardship behaviors [44].

Animal-assisted interventions show promise for mood and social engagement, but effect sizes vary and protocols must protect animal welfare. Educational programs that integrate live observation with structured reflection (what cues did you use to infer the animal’s state?) improve empathic accuracy and ethical reasoning. Virtual and augmented reality offer safe exposure to less familiar taxa, allowing for controlled manipulation of cues (e.g., enhancing eye visibility or translating chemical signals into visual proxies) to test how recognizability shapes empathy.

Contemporary models describe mind perception along two dimensions: agency (capacity for planning and self-control) and experience (capacity to feel pain and pleasure). Mammals are granted both by default; reptiles and insects are often assigned low agency and ambiguous experience. The gradient of empathy can be interpreted as a function of the prior probability that an observer assigns to these dimensions. Demonstrations of goal pursuit, tool use, vocal learning, parental care, or social grooming shift beliefs upward for distant taxa, increasing moral concern without changing phylogeny.

Design elements can further modulate engagement. Visual representations that incorporate face-like reference points, particularly eyes and gaze cues, exploit evolved mechanisms of attention and empathy, thereby enhancing emotional connection to otherwise neglected taxa [10]. Narrative framing is equally critical: when media portrayals of reptiles or invertebrates avoid sensationalist tropes of danger and instead emphasize agency, ecological roles, or problem-solving capacities, observers report higher perceived sentience and reduced fear [45].

Zoos and aquaria represent particularly fertile contexts for intervention. Positioning non-mammalian exhibits adjacent to interactive cognition displays—for example, problem-solving puzzles for octopuses or evidence of tool use in corvids—can elevate perceived intelligence and moral concern [46,47]. Such strategies not only broaden the scope of empathy but also promote a more equitable distribution of conservation concern, challenging entrenched charisma biases.

Although zoos and aquaria have the potential to act as powerful platforms for reshaping empathy toward non-mammalian taxa, in practice they rarely exploit this opportunity. Several structural and cognitive barriers explain this shortfall.

First, institutional priorities are heavily skewed toward charismatic megafauna—large mammals such as elephants, primates, and big cats—which attract more visitors and funding [48]. Exhibits for reptiles, amphibians, fish, or invertebrates are often underfunded, relegated to peripheral spaces, and presented with minimal interpretive material. This reinforces the “charisma bias” in public attention.

Second, exhibit design tends to emphasize passive display rather than interactive cognitive engagement. Non-mammalian species are frequently housed in static enclosures with limited enrichment. Without visible problem-solving, play, or social communication, visitors receive few cues that would invite attribution of sentience or agency [49].

Third, communication strategies often perpetuate negative or neutral framings. Snakes are described in terms of danger, insects as pests, and fish as decorative. Even when ecological importance is mentioned, it is rarely translated into narratives that resonate with human social cognition—such as parental care, cooperation, or exploration [50].

Fourth, resource allocation and staff expertise limit innovation. Designing cognitive displays for octopuses or problem-solving experiments for crows requires ethological expertise and specialized infrastructure, which many institutions lack or deprioritize.

Finally, visitor expectations constrain programming. Many zoo-goers seek entertainment rather than educational transformation, which incentivizes institutions to invest in photogenic mammals rather than cognitively demanding exhibits for “less-loved” taxa [51].

In sum, while zoos possess unique potential to broaden empathy beyond mammals, prevailing economic, cultural, and design logics reproduce rather than challenge anthropocentric biases. Reorienting zoo education toward cognitive transparency, ecological framing, and narrative reframing could unlock this capacity, but would require a fundamental institutional shift.

For practitioners—teachers, conservationists, communicators—we suggest: (1) Start with mammals to build confidence, then bridge to distant taxa via shared features (parental care, play, problem solving); (2) Use high-quality visuals that foreground eyes, motion, and context; (3) Pair facts with short narratives about individual animals to enhance person-specific empathy; (4) Scaffold hands-on, ethical encounters where feasible; (5) Evaluate interventions with pre/post measures and iterate; (6) Collaborate with local cultural leaders to align messaging with community values.

**Table 1 biology-14-01438-t001:** Interventions that Reshape Empathy Toward Non-Mammalian Species.

Type of Intervention	Mechanisms of Action	Observed Outcomes	Representative References
**Educational Modules (classroom, outreach, short workshops)**	Present factual knowledge about ecological and intrinsic value of reptiles, fish, or invertebrates; emphasize functional roles (e.g., pollination, pest control, nutrient cycling); use visual analogies with mammalian care.	Increased willingness to protect non-mammalian taxa; measurable shifts in conservation attitudes even after brief exposure; improved retention of ecological knowledge.	[15,44]
**Citizen-Science Programs**	Direct, hands-on interaction (handling, observing, sampling); co-creation of data; fostering sense of agency and contribution.	Reduced disgust and fear toward reptiles, amphibians, arthropods; heightened stewardship, environmental identity, and empathy toward local biodiversity.	[52]
**Visual Design Enhancements**	Use of eye-spots, gaze cues, or anthropomorphic cues to create perceptual anchors; emphasize “faces” or recognizable body plans.	Increased engagement, attention, and likability of animals that otherwise evoke low empathy (e.g., fish, insects).	[53]
**Narrative Reframing**	Avoid sensationalist tropes (e.g., “dangerous snakes”); instead highlight cooperation, ecological services, parental care, and survival strategies.	Reduced fear-based stereotypes; greater acceptance of coexistence with species traditionally stigmatized.	[51]
**Zoo & Aquarium Cognitive Displays**	Place non-mammalian taxa in interactive contexts (e.g., octopuses solving puzzles, fish navigating mazes); emphasize cognition, learning, and social communication.	Elevates perceived sentience, agency, and moral concern; expands conservation support beyond mammals.	[54]
**Virtual & Augmented Reality Experiences**	Immersive simulations of animal perspective or habitat; promote embodiment and perspective-taking.	Strong increases in empathy, conservation donations, and prosocial intentions toward marine invertebrates and fish.	[55]

## 16. Methodological Caveats and Confounds

Comparative empathy research faces pitfalls. Stimuli are often unmatched (high-quality mammal photos vs. low-quality invertebrate images). Familiarity, danger, cultural narratives, and aesthetic preferences confound phylogenetic distance. Cross-cultural replication is still limited, and WEIRD samples dominate. Many measures infer empathy from self-report rather than behavior. Ethically, research must avoid harm to animals used as stimuli. Future work should use preregistered designs, standardized stimulus sets, and mixed methods (behavioral + physiological + neural) with adequate power and cultural diversity.

## 17. Open Questions and Research Agenda for Future Directions

### 17.1. Lifespan Plasticity

The empathy gradient is likely plastic rather than fixed. Early biases toward face-like cues, biological motion, and gaze following emerge in infancy, scaffolding later social learning; yet contact and instruction can reshape category boundaries across development. Repeated, meaningful exposure—from early childhood nature experiences to structured programs in adolescence—predicts stronger concern for non-mammalian taxa, counteracting the “extinction of experience” in urban contexts [56,57].

Classic contact effects generalize: positive, cooperative encounters reduce anxiety and increase inclusion of the “other”, suggesting that guided contact with distant taxa (e.g., amphibians, insects) should broaden moral regard without diminishing care for mammals [58].

### 17.2. Which Experiences Work Best?

Brief instructional modules that explain ecological function and intrinsic value can shift stated preferences and willingness to protect within a session; citizen-science and hands-on engagement produce larger, more durable effects by coupling knowledge with agency [44,52]. Design choices matter: eye-like reference points and clear action cues improve perceptual readability, while reframing narratives away from threat tropes reduces disgust and fear.

### 17.3. Neuroendocrine Leverage—Opportunities and Limits

Naturalistic pathways (mutual gaze, affiliative touch) recruit oxytocinergic and reward circuitry during human–animal interaction [4,28]. However, proposals to use exogenous oxytocin (or dopaminergic agents) in educational settings raise safety, efficacy, and ethical concerns: pharmacokinetics are debated, and effects are heterogeneous across individuals and contexts [59]. A prudent approach is to mimic endogenous triggers—structured, positive, controllable contact—rather than administer hormones, while quantifying outcomes via registered trials.

### 17.4. Incorporating Public Empathy into Conservation Triage

Empathy can increase compliance, donations, and stewardship, but unfiltered charisma bias risks misallocating scarce funds toward already popular mammals. Decision frameworks should integrate sentience, suffering capacity, and ecological function, using transparent multi-criteria methods to align public values with ecosystem needs [60,61]. Rather than capitulating to charisma, targeted communication can highlight agency, problem solving, parental care, or ecosystem services in under-loved taxa to ethically broaden concern.

### 17.5. Computational Prediction and Cross-Cultural Validation

A constructive goal is to build predictive models that map species features (e.g., perceived agency/experience, cue readability, ecological role, risk signals) onto moral concern and policy support. Foundational work on mind perception and anthropomorphism provides candidate feature spaces [61,62]. Models should be neurally informed (linking predictions to empathy-related circuits [63] and evaluated across cultures to avoid WEIRD sampling artifacts [64]. Convergence across longitudinal fieldwork, controlled interventions, cross-cultural surveys, and model-based neuroscience will be critical for answering these questions with causal traction.

Legal protections for animals consistently mirror gradients of public empathy. In most jurisdictions, vertebrates—and particularly mammals—receive the strongest baseline protections, while invertebrates are often excluded or only weakly covered under welfare legislation. This anthropocentric asymmetry reflects both intuitive empathy gradients and historical inertia in policy-making [65]. Yet, scientific advances increasingly challenge these defaults. Experimental evidence now supports the capacity for nociception, associative learning, and even forms of problem-solving in several invertebrate clades, most prominently cephalopods.

Refs. [54,66] and decapod crustaceans [67]. Recognition of such findings has already reshaped policy landscapes: the European Union and the United Kingdom have formally incorporated cephalopods and decapod crustaceans into their animal welfare frameworks, signaling a paradigm shift away from protections based solely on phylogenetic proximity [68].

A science-based approach to welfare legislation would therefore move beyond charisma and public preference, adopting evidence-based criteria such as demonstrated nociception, learning ability, and behavioral flexibility [69] (Figure 4). Importantly, extending minimal protections to phylogenetically distant taxa need not dilute robust protections already secured for mammals. Instead, it raises the moral and legal floor while preserving high ceilings where strong cultural and political support already exists. Such a strategy reflects an incremental but cumulative broadening of the “circle of moral concern” [70], aligning policy with empirical evidence and ethical reasoning.

Ultimately, the expansion of welfare protections in line with scientific evidence does more than remedy inconsistencies: it acknowledges that sentience and the capacity to suffer are distributed across evolutionary space. In doing so, legislation can better reflect both moral responsibility and One Health imperatives, recognizing that ecological stability, human well-being, and animal welfare are deeply interdependent.

Empathy gradients shape policy. Charismatic mammals dominate conservation marketing, leaving amphibians, reptiles, and invertebrates relatively underfunded despite steep declines and critical ecosystem services. Messaging experiments show that emphasizing functional importance (e.g., insect pollination, amphibian pest control), parental care, or cognitive complexity can narrow funding gaps. Images that provide scale, eyes, and social context outperform abstract habitat photos. Partnerships with local communities that benefit directly from ecosystem services sustain long-term support.

## 18. Conclusions

Our synthesis risks anthropocentrism by centering human perceptual systems. A fuller account would also consider how animals perceive humans—many species treat us as predators or conspecifics—and how bidirectional signaling shapes relationships. Additionally, phylogenetic proximity is entangled with exposure; urban dwellers meet dogs daily but rarely interact with amphibians [70]. Untangling causal threads requires longitudinal designs and natural experiments where exposure changes (e.g., after an educational program or a move to a biodiverse region).

Anthropomorphism—projecting human-like traits onto nonhumans—can aid learning when used judiciously, because it bridges unfamiliar signals to familiar schemas. Yet, it risks misrepresenting species-specific needs (e.g., assuming solitary reptiles crave constant handling). A scientifically responsible approach uses ‘calibrated anthropomorphism’: leverage accessible metaphors for agency and affect while explicitly teaching where the analogy breaks down. This strategy keeps empathy inclusive without erasing difference.

Humans’ preferential concern for phylogenetically close species reflects the interplay of perceptual attunement, neuroendocrine reinforcement, and cultural narratives that valorize mammalian companionship. Yet, the empathic resonance with cephalopods, cetaceans, and parrots demonstrates that evolutionary distance does not fix moral boundaries: features that enable mind attribution and social prediction can bridge even wide phylogenetic divides. A scientifically grounded understanding of these mechanisms empowers educators, ethicists, and conservationists to deliberately widen the moral circle (Figure 5). Here, the One Health framework [71] offers a unifying paradigm: by recognizing that the well-being of humans, animals, and ecosystems is inextricably interdependent, it reframes empathy not as a scarce resource to be rationed, but as a scalable principle guiding inclusive stewardship. The challenge ahead is not to deny evolved human affinities but to channel them—through education, ethical design, and conservation policy—toward a vision of coexistence in which protecting biodiversity is understood as protecting ourselves. If empathy can be extended beyond its ancestral anchors, conservation will no longer depend on charisma, but on an enlightened recognition that all species are threads in the same web of planetary health.

## Figures and Tables

**Figure 1 biology-14-01438-f001:**
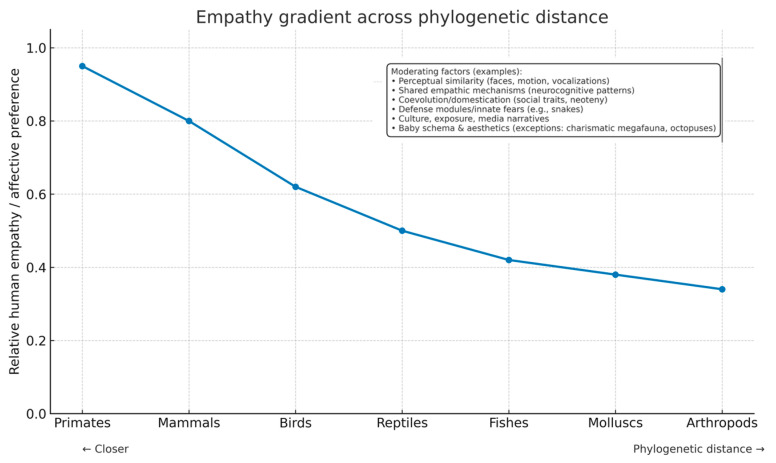
Relative Human empathy decreases as phylogenetic distance increases (conceptual trend). Representative clades are shown on the x-axis; moderators discussed in text include perceptual similarity, predictive social cognition, neuroendocrine modulation, domestication, and cultural exposure.

**Figure 2 biology-14-01438-f002:**
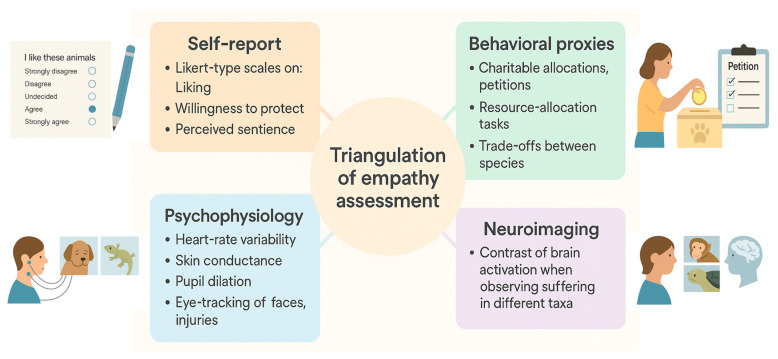
Triangulation of empathy assessment across multiple methodologies. Self-reports capture explicit attitudes (liking, protection, perceived sentience), while behavioral proxies index actual commitments (donations, petitions, allocation tasks). Psychophysiology records embodied reactivity (heart-rate variability, conductance, pupil dilation, gaze), and neuroimaging contrasts brain activation to animal suffering across taxa. Together these methods provide convergent evidence for a graded yet flexible structure of human–animal empathy.

**Figure 3 biology-14-01438-f003:**
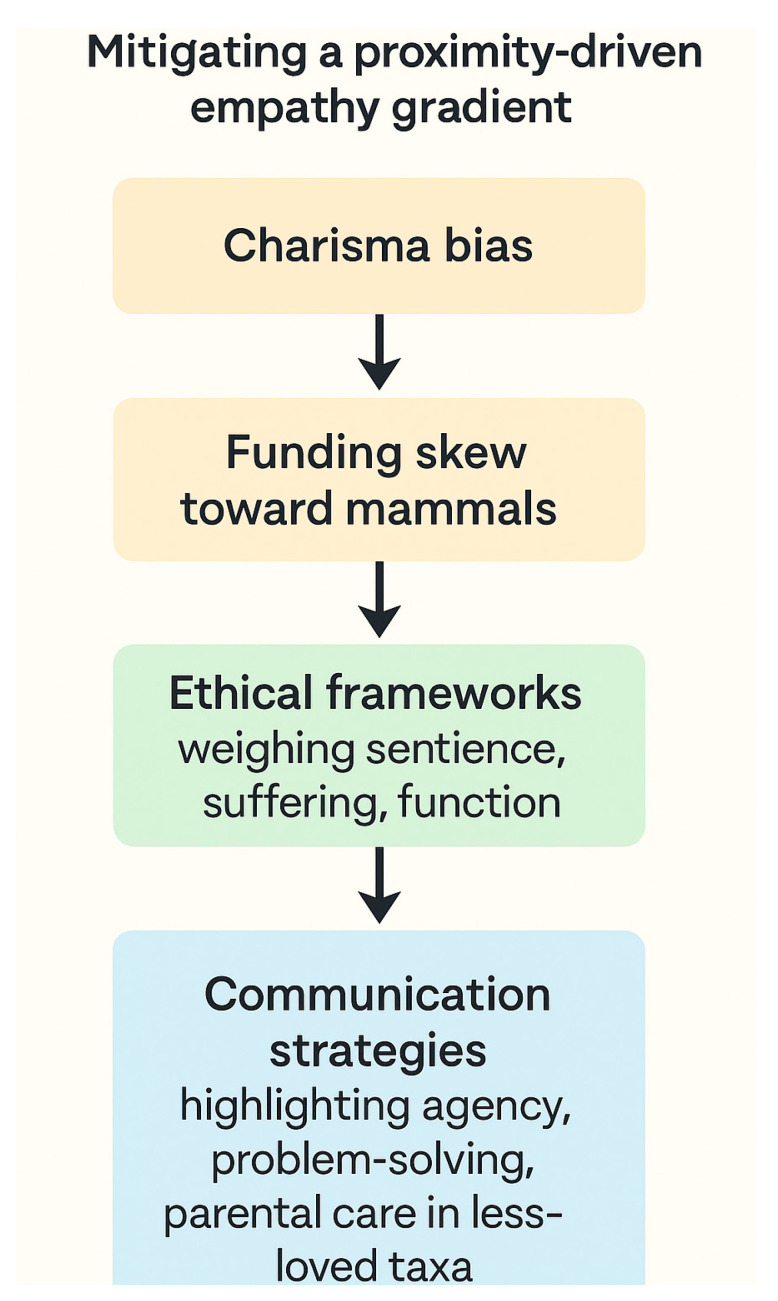
Charisma Bias in Conservation Decision-Making. This infographic illustrates the phenomenon of charisma bias, where public empathy and conservation funding disproportionately favor mammals and birds, while ecologically critical but less charismatic taxa (e.g., insects, amphibians, detritivores) receive less attention. Ethical frameworks that incorporate sentience, suffering capacity, and ecological function can counterbalance anthropocentric biases. Communication strategies emphasizing agency, problem-solving abilities, and parental care in underappreciated species provide practical tools to broaden public concern and foster more equitable biodiversity protection.

**Figure 4 biology-14-01438-f004:**
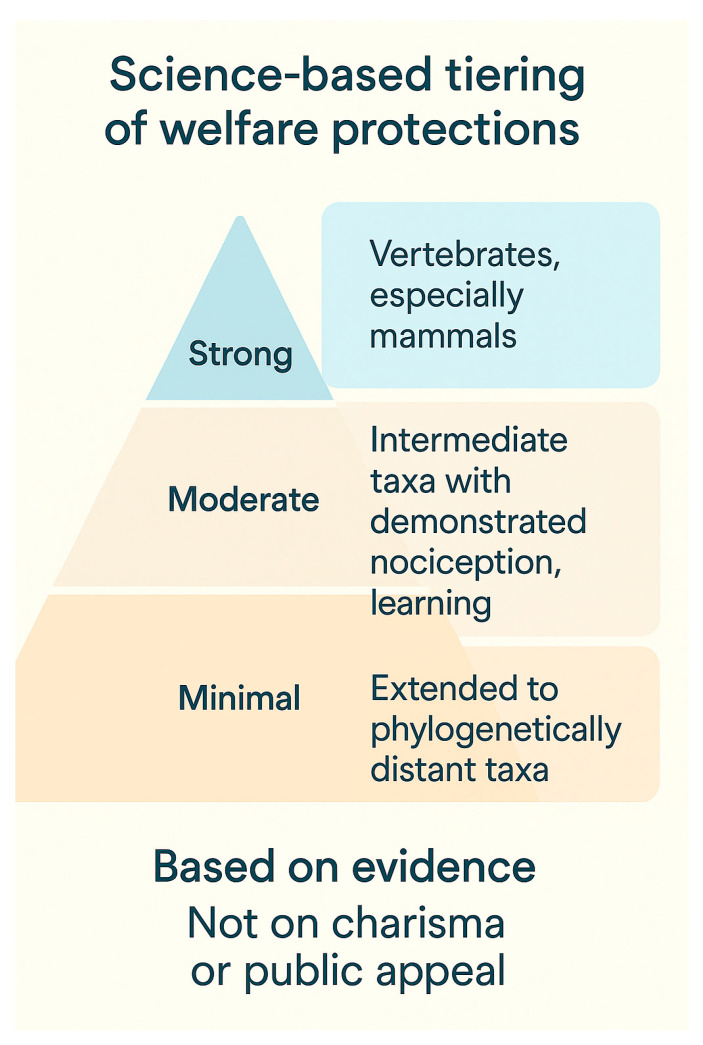
A science-based approach to welfare legislation would therefore move beyond charisma and public preference, adopting evidence-based criteria.

**Figure 5 biology-14-01438-f005:**
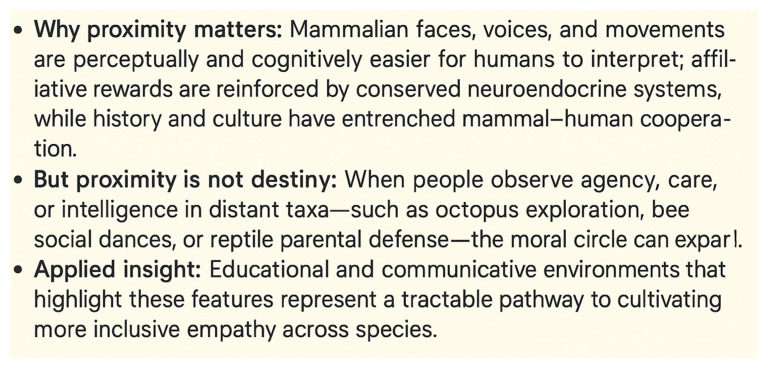
Proximity and expanding the moral circle.

## Data Availability

No new data were created or analyzed in this study. Data sharing is not applicable.

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
