# Peer review of "Why Humans Prefer Phylogenetically Closer Species: An Evolutionary, Neurocognitive, and Cultural Synthesis"

_biology, 2025, doi:10.3390/biology14101438_

Round 1
Reviewer 1 Report
Comments and Suggestions for Authors
I have reviewed this piece of research and in my opinion several points should be addressed before further consideration.
My specific comments are listed below:
1. Abstract is vague without reporting background, objectives, and major findings of the study.
2. Introduction is poorly written without detailing the background studies and latest research progress in human choice and preferences.
This study also lacks the research gaps still exist.
3. The evolution of proximity hypothesis should be added with appropriate citations.
4. How do domestication and convolution affects preferred species? Please provide adequate details with supporting references.
5. State the role of Ethical Implications and Conservation with sufficient details.
6. Lifespan plasticity. Please give more emphasis in this section.
7. Conclusion needs to report limitations of this study and reasearch gaps that persists.
Author Response
Step-by-step response to rev. 1 for biology I sincerely thank the reviewers and the handling editor for their thoughtful and constructive comments, which have greatly improved the clarity, scientific depth, and interdisciplinary coherence of this paper. Below we provide a detailed point-by-point response. Rev. 1 1. Abstract is vague without reporting background, objectives, and major findings of the study. I rewrote the abstract 2. Introduction is poorly written without detailing the background studies and latest research progress in human choice and preferences. This study also lacks the research gaps still exist I rewrote the introduction and changed the title of the figure. 3. The evolution of proximity hypothesis should be added with appropriate citations. I have done this task 4. How do domestication and convolution affects preferred species? Please provide adequate details with supporting references. I changed the title to paragraph 4 and added details on domestication etc. 5. State the role of Ethical Implications and Conservation with sufficient details. I added details 6. Lifespan plasticity. Please give more emphasis in this section. I expanded this paragraph 7. Conclusion needs to report limitations of this study and reasearch gaps that persists. I added this in the conclusions.
Reviewer 2 Report
Comments and Suggestions for Authors
This is a very important and much needed discussion of the connection between humans and other animals (or sometimes why there is a disconnect). This essay will add to the literature and I'm glad to see it taking place outside of animal media studies. What i didn't see was the use of the term "neoteny," when the author was discussing large features such as eyes and ears and what stimulates in humans. Also, define empathy more deeply. I've inserted a number of comments/questions/suggestions into the document. Importantly, there is a large literature on media portrayals that should be considered in this section. I've written more on this in the note for that section. The author should further expand the One Health framework to include work by Ferdowsian and others on a Just One Health view, that regards health as a right. This was an interesting read that spans several disciplines and is well written.
For more comments, please see the attachment

Author Response
Rev. 2 I sincerely thank the reviewers and the handling editor for their thoughtful and constructive comments, which have greatly improved the clarity, scientific depth, and interdisciplinary coherence of this paper. Below we provide a detailed point-by-point response. This is a very important and much needed discussion of the connection between humans and other animals (or sometimes why there is a disconnect). This essay will add to the literature and I'm glad to see it taking place outside of animal media studies. What i didn't see was the use of the term "neoteny," when the author was discussing large features such as eyes and ears and what stimulates in humans. Also, define empathy more deeply. I've inserted a number of comments/questions/suggestions into the document. Importantly, there is a large literature on media portrayals that should be considered in this section. I have completely changed paragraph 10 to include these important concepts I've written more on this in the note for that section. The author should further expand the One Health framework to include work by Ferdowsian and others on a Just One Health view, that regards health as a right. This was an interesting read that spans several disciplines and is well written. I have added paragraph 10a to expand on this subject. I also added table S1 to improve understanding of the added material.
Round 2
Reviewer 1 Report
Comments and Suggestions for Authors
Manuscript has been improved.